# PeerJ

# The role of a water bug, *Sigara striata*, in freshwater food webs

Jan Klecka

Laboratory of Theoretical Ecology, Biology Centre of the Academy of Sciences of the Czech
Republic, Institute of Entomology, České Budějovice, Czech Republic
Department of Fish Ecology and Evolution, EAWAG, Swiss Federal Institute of Aquatic Science
and Technology, Kastanienbaum, Switzerland

## ABSTRACT

Freshwater food webs are dominated by aquatic invertebrates whose trophic relation-
ships are often poorly known. Here, I used laboratory experiments to study the role
of a water bug, *Sigara striata*, as a potential predator and prey in food webs of stag-
nant waters. Multiple-choice predation experiment revealed that *Sigara*, which had
been considered mostly herbivorous, also consumed larvae of *Chironomus* midges.
Because they often occur in high densities and are among the most ubiquitous
aquatic insects, *Sigara* water bugs may be important predators in fresh waters. A
second experiment tested the role of *Sigara* as a potential prey for 13 common inver-
tebrate predators. Mortality of *Sigara* inflicted by different predators varied widely,
especially depending on body mass, foraging mode (ambush/searching) and feeding
mode (chewing/suctorial) of the predators. *Sigara* was highly vulnerable to ambush
predators, while searching predators caused on average 8.1 times lower mortality
of *Sigara*. Additionally, suctorial predators consumed on average 6.6 times more
*Sigara* individuals than chewing predators, which supports previous results hinting
on potentially different predation pressures of these two types of predators on prey
populations. The importance of these two foraging-related traits demonstrates the
need to move from body mass based to multiple trait based descriptions of food web
structure. Overall, the results suggests that detailed experimental studies of common
but insufficiently known species can significantly enhance our understanding of food
web structure.

Corresponding author
Jan Klecka, jan.klecka@entu.cas.cz

**Subjects** Aquaculture, Fisheries and Fish Science, Ecology, Entomology, Environmental Sciences,
Zoology
**Keywords** Predation, Predator–prey interactions, Food webs, Foraging, Heteroptera, Corixidae

## INTRODUCTION

The view of ecological communities as networks of interacting species has revolutionized
research of community structure, stability and responses to environmental changes (*Ings et
al., 2009*). Studies that combine modelling and field data are attempting to provide general
explanations of mechanisms structuring natural communities (*Bascompte et al., 2003*;
*Beckerman, Petchey & Warren, 2006*; *Bascompte & Jordano, 2007*; *Petchey et al., 2008*) and
to predict their responses to various threats, such as climate change (*Petchey, Brose & Rall,
2010*; *O'Gorman et al., 2012*) and habitat destruction (*Melián & Bascompte, 2002*; *Fortuna
& Bascompte, 2006*). Having accurate and detailed information about trophic interactions

of individual species forming food webs is a necessary condition for these attempts to succeed. Although the resolution of published food webs has increased considerably in recent years (*Thompson, Dunne & Woodward, 2012*), trophic position of many common species remains uncertain. This is troubling because a vast body of theoretical research has shown that food web structure is a key to understanding food web dynamics (*de Ruiter, Neutel & Moore, 1995*; *McCann, Hastings & Huxel, 1998*). Limited knowledge of food web structure in individual habitats thus calls into question the reliability of predictions of consequences of climate change or habitat destruction on food web diversity and stability.

Freshwater food webs are dominated by invertebrates, such as adults and larvae of insects, many of them carnivorous. Predatory aquatic insects have been traditionally considered generalists (*Peckarsky, 1982*) and food web studies often include them in relatively low resolution. This approach runs the risk of missing many crucial details given the tremendous diversity of aquatic insects. For example, there is over 150 species of diving beetles (all of them predatory) in the Czech Republic (*Boukal et al., 2007*), where this study has been conducted. Tens of species can coexist locally (*Klecka & Boukal, 2011*), possibly thanks to pronounced prey selectivity (*Klecka & Boukal, 2012*). Selective feeding in predatory aquatic insects is characterized not only by interspecific differences in prey selectivity and diet breadth (*Cooper, Smith & Bence, 1985*; *Allan, Flecker & McClintock, 1987*; *Culler & Lamp, 2009*; *Klecka & Boukal, 2012*), but also by marked ontogenetic diet shifts (*Woodward & Hildrew, 2002*; *Klecka & Boukal, 2012*). Prey mortality results from the interaction of several key predator and prey traits, particularly body size, predator foraging behaviour and prey vulnerability traits, such as its ability of rapid escape (*Klecka & Boukal, 2013*). The insect component of freshwater food webs is thus complex, but we are beginning to understand mechanisms structuring its interactions. The importance of predatory insects is underscored by findings of trophic cascades elicited by individual species, such as *Dytiscus alascanus* (*Cobbaert, Bayley & Greter, 2010*) and *Notonecta* sp. (*Arnér et al., 1998*).

Although detailed laboratory studies have been performed with a range of species, descriptions of food web structure have relied mostly on gut contents analyses of field-collected specimens (*Warren, 1989*; *Woodward & Hildrew, 2002*; *Layer et al., 2010*). The most serious downside of this approach is that it cannot be reliably used to study the diet of suctorial predators, such as water bugs (Heteroptera). Even recent studies rely on expert knowledge or previous literature data to infer trophic relationships of suctorial predators (e.g., *Layer et al., 2010*). This way of circumventing the problem of identifying prey of suctorial predators may be dangerous. Feeding relationships of most species are virtually unknown and existing literature provides conflicting information in many other cases, including a decades-long controversy about herbivorous or predatory nature of corixid bugs (Heteroptera: Corixidae) (*Hutchinson, 1993*; *Popham, Bryant & Savage, 1984*) studied in this paper. Despite several early attempts at identification of prey remains in the gut of suctorial water bugs by techniques of molecular biology (*Giller, 1982*; *Giller, 1984*; *Giller, 1986*), little progress has been achieved (*Tate & Hershey, 2003*; *Morales et al., 2003*). Modern molecular assays detecting prey DNA in the gut using species-specific

markers have not yet been widely employed in freshwater predators, although they are increasingly used in terrestrial invertebrates (*Günther et al., in press*; *Foltan et al., 2005*; *Symondson, 2002*) and hold a significant promise for the future (*Pompanon et al., 2012*). In this situation, traditional laboratory experiments still offer a unique opportunity to obtain reliable high-resolution data on predator–prey interactions of individual species in freshwater food webs.

The aim of this study was to evaluate the role of a common freshwater insect, *Sigara striata*, in freshwater food webs. *Sigara* and other corixid water bugs are among the most ubiquitous and abundant aquatic insects (*Schilling, Loftin & Huryn, 2009*) and often reach densities of tens (*Tolonen et al., 2003*) or even hundreds of individuals m$^{-2}$ (*Bendell & McNicol, 1995*), but their trophic position is not well understood (*Hutchinson, 1993*). I performed two complementary experiments addressing this knowledge gap. First, I tested whether *Sigara* can consume seven locally abundant species of aquatic invertebrates. Second, I tested the mortality of *Sigara* water bugs caused by 13 common predatory insects (nine species, multiple life stages in three cases). The same set of species, both predators and prey, as in *Klecka & Boukal (2012)* and *Klecka & Boukal (2013)* was used. The first experiment addressed a decades-old controversy concerning whether *Sigara* is strictly herbivorous (scraping algae from submerged vegetation, stones etc.) or whether it also feeds on other invertebrates. If carnivorous, *Sigara* could be an important freshwater predator because of its frequently very high population density.

## METHODS

### Experiments

The water bug *Sigara striata* (Heteroptera: Corixidae) was collected in small pools in a reclaimed sandpit near Suchdol nad Lužnicí in South Bohemia, Czech Republic. Some of its potential predators and preys were collected at the same site and others in various small fishless water bodies close to the city of České Budějovice. Species were selected to represent a wide variety of regionally dominant species and to form a taxonomically and functionally diverse assemblage. Experiments were carried out in May and June 2007 in a climate room with a regular temperature cycle (day: max. 22 °C, night: min. 18 °C; mean 20 °C) and 18L:6D photoperiod. The experimental procedures followed *Klecka & Boukal (2012)* and *Klecka & Boukal (2013)* and the same species of predators and prey, except for *Sigara*, were used. All animals were kept in the lab for 2–5 days prior to the experiments to allow for acclimation to the laboratory conditions. Predators were fed daily ad libitum with prey different from *Sigara* (mainly larvae of Trichoptera) and starved for 24 h prior to the experiment. Experiments were performed in plastic boxes filled with 2.5 l of aged tap water (bottom dimensions 24 × 16 cm, water depth ca. 8 cm). The vessels had no substrate on the bottom but contained simple perching sites formed by four stripes of white plastic mesh suspended vertically in the water column. The vessels were shielded by sheets of brown carton from all sides to prevent disturbance of the experiments.

The first experiment tested whether *Sigara striata* can feed on six species of invertebrates which co-occur with it frequently in natural habitats (Table 1). In each replicate, six

**Table 1 The list of species used as potential prey of *Sigara striata* and their traits.** The same set of species but different individuals was used by *Klecka & Boukal (2012)* and *Klecka & Boukal (2013)*.

| Species | N | Body length (mm) | | Body mass (mg) | | Microhabitat |
|---|---|---|---|---|---|---|
| | | Mean | SD | Mean | SD | |
| *Asellus aquaticus* adult | 10 | 7.21 | 0.99 | 1.69 | 0.38 | benthic |
| *Chironomus* sp. larva | 10 | 9.12 | 0.71 | 0.31 | 0.069 | benthic |
| *Cloeon dipterum* larva | 10 | 6.87 | 0.89 | 1.02 | 0.21 | benthic |
| *Culex* sp. larva | 10 | 8.92 | 0.41 | 0.62 | 0.17 | pelagic |
| *Daphnia* sp. adult | 20 | 2.21 | 0.19 | 0.041 | 0.029 | pelagic |
| *Lymnaea stagnalis* juvenile | 10 | 9.65[a] | 0.77 | 7.84[b] | 2.01 | pelagic[c] |

**Notes.**

*N*, number of individuals measured and weighed.

[a] Shell length measured.

[b] Weighed without shell.

[c] *Lymnaea* was crawling on the sides of the experimental vessels and on the suspended mesh.

juvenile *Lymnaea* snails, 10 *Chironomus* midge larvae, 10 *Cloeon* mayfly larvae, 10 *Culex* mosquito larvae, 10 adult *Asellus* isopods and 30 adult *Daphnia* cladocerans were introduced into a vessel with water (see above) and 10 *Sigara* individuals were added after ca. 10 min. Ten *Sigara* individuals were used to increase the chance of detecting predation even if it occurs only rarely. Six replicates with *Sigara* as a potential predator and six without *Sigara* were performed to compare prey mortality in the presence/absence of *Sigara*. Qualitative observations of *Sigara* and prey behaviour were conducted in the beginning of the experiment and than occasionally during the experiment. All *Sigara* individuals used in the experiment and a sample of all prey species were preserved in 80% ethanol, their body length excluding appendages was measured to nearest 0.1 mm and their body mass was weighed after 48 h of drying at 50 °C.

The second experiment aimed to estimate mortality rate of *Sigara* inflicted by different predators (Table 2). In each replicate, ten adults of *Sigara* were released first and one predator was added after ca. 10 min. The number of surviving *Sigara* individuals was counted after 24 h. All individuals of *Sigara* and the predators were used only once. Four control trials were run to evaluate natural mortality of *Sigara*; no individual died in any of these control trials, suggesting that mortality observed in the presence of predators was caused entirely by predation. All predators were preserved in 80% ethanol, their body length measured and their body mass weighed as in the first experiment. I also classified their microhabitat use and made qualitative observations of their behaviour during the experiments.

## Data analysis

I used Bayesian methods to estimate predation rates of *Sigara* consuming other invertebrates in the first experiment. The same approach was used to estimate mortality rate of *Sigara* caused by different predators and to test the role of predator traits for *Sigara* mortality in the second experiment. Overdispersed binomial distribution and logistic link function was used in all cases.

**Table 2 The list of predators used in the experiment and their traits.** The same set of species but different individuals was used by *Klecka & Boukal (2012)* and *Klecka & Boukal (2013)*.

| Species | N | Body length (mm) | | Body mass (mg) | | Foraging mode | Feeding mode | Microhabitat |
|---|---|---|---|---|---|---|---|---|
| | | Mean | SD | Mean | SD | | | |
| Coleoptera | | | | | | | | |
| *Acilius canaliculatus* adult | 4 | 15.6 | 0.57 | 61.74 | 9.34 | searching | chewing | benthic |
| *Acilius canaliculatus* L2 | 4 | 11.8 | 0.63 | 2.74 | 0.66 | ambush | suctorial | pelagic |
| *Acilius canaliculatus* L3 | 4 | 22.2 | 1.75 | 14.66 | 4.70 | ambush | suctorial | pelagic |
| *Dytiscus marginalis* adult | 4 | 31.7 | 0.89 | 528.43 | 50.88 | searching | chewing | benthic |
| *Dytiscus marginalis* L3 | 4 | 49.1 | 3.01 | 176.43 | 76.12 | ambush | suctorial | pelagic |
| *Hydaticus seminiger* adult | 4 | 14.5 | 0.32 | 64.94 | 8.42 | searching | chewing | benthic |
| Hemiptera | | | | | | | | |
| *Ilyocoris cimicoides* adult | 4 | 13.9 | 0.60 | 34.43 | 6.85 | searching | suctorial | benthic |
| *Notonecta glauca* adult | 5 | 15.3 | 0.44 | 39.43 | 8.08 | ambush | suctorial | pelagic |
| Odonata | | | | | | | | |
| *Anax imperator* F-0 | 4 | 47.6 | 2.61 | 267.0 | 54.42 | ambush | chewing | pelagic |
| *Coenagrion puella* F-0 | 4 | 12.8 | 0.87 | 4.80 | 0.99 | ambush | chewing | pelagic |
| *Libellula depressa* F-0 | 5 | 22.3 | 1.11 | 58.41 | 19.51 | ambush | chewing | benthic |
| *Libellula depressa* F-2 | 4 | 15.7 | 0.72 | 20.94 | 5.57 | ambush | chewing | benthic |
| *Sympetrum sanguineum* F-0 | 4 | 16.1 | 0.97 | 20.82 | 4.49 | ambush | chewing | pelagic |

**Notes.**
L2, larvae of the second instar; L3, larvae of the third instar; F-0, larvae of the last instar; F-2, larvae of the second before the last instar; $N$, number of replicates.

To estimate predation rates of *Sigara* on different prey species, I used a model contrasting mortality rate of prey species $i$ in the presence of *Sigara* and in control trials without *Sigara*. This approach was needed because some prey species had non-zero mortality in the control trials. I assumed that mortality $m_i$ of prey $i$ depends on the presence of *Sigara* in a species-specific way. To account for overdispersion, I used a two-stage model:

$$Y_i \sim Bin(n_i, m_i)$$
$$logit(m_i) = a_i + b_i S_i + z_i \tag{1}$$
$$z_i \sim N(0, \sigma^2)$$

where $Y_i$ is the number of individuals of a prey species $i$ dying in an experiment, $n_i$ is the initial number of prey individuals, $m_i$ is the mortality of prey $i$, $a_i$ is the natural mortality of prey $i$ in the absence of *Sigara*, $b_i$ is the effect of the presence of *Sigara* on the mortality of prey $i$, $S_i$ denotes the presence of *Sigara* ($S_i = 1$ when *Sigara* was present, $S_i = 0$ otherwise), and $z_i$ is the overdispersion term.

To estimate the mortality rate of *Sigara* caused by individual predators, I used a model where the mortality of *Sigara* is a function of predator species. Predator species was used as a random factor. Two-stage model accounting for overdispersion was used similarly as above:

$$Y_i \sim Bin(n_i, m_i)$$
$$logit(m_i) = a_0 + b_i + z_i \tag{2}$$
$$b_i \sim N(0, \sigma_1^2)$$
$$z_i \sim N(0, \sigma_2^2)$$

where $Y_i$ is the number of *Sigara* individuals dying in an experiment, $n_i$ is the initial number of individuals, $m_i$ is the mortality of *Sigara*, $a_0$ is the intercept, $b_i$ is the effect of predator $i$ on the mortality of *Sigara* and $z_i$ is the overdispersion term.

To estimate the dependence of *Sigara* mortality on predator traits, I used a hierarchical model where the mortality of *Sigara* $m_i$ is a function of predator species (a random factor) as in Eq. (2). The mortality of *Sigara* further depends on a combination of several predator traits with additive effects at the scale of the linear predictor. Two-stage model accounting for overdispersion was used as above; linear predictor in Eq. (3) contains all explanatory variables:

$$Y_i \sim Bin(n_i, m_i)$$
$$logit(m_i) = a_0 + a_1(ln(w_i) - ln(\bar{w})) + a_2(ln(w_i) - ln(\bar{w}))^2$$
$$+ a_3 A_i + a_4 B_i + a_5 C_i + b_i + z_i \tag{3}$$
$$b_i \sim N(0, \sigma_1^2)$$
$$z_i \sim N(0, \sigma_2^2)$$

where $Y_i$ is the number of *Sigara* individuals dying in an experiment, $n_i$ is the initial number of individuals, $m_i$ is the mortality of *Sigara*, $a_0$ is the intercept, $a_1, \ldots, a_5$ are parameters describing the effect of individual predator traits, $w_i$ is body mass of predator $i$, $\bar{w}$ is mean body mass of all predators (i.e., predator body mass is centered in Eq. (3)), $A_i$ is the foraging mode of predator $i$ ($A_i = 1$ for ambush predators and $A_i = 0$ for searching predators), $B_i$ is the feeding mode ($B_i = 1$ for ambush predators and $B_i = 0$ for searching predators), $C_i$ is the microhabitat preference of predator $i$ ($C_i = 1$ for ambush predators and $C_i = 0$ for searching predators) and $z_i$ is the overdispersion term. The analysis of posterior parameter distributions revealed that parameters $a_2$ and $a_5$ were approximately zero; the corresponding terms in Eq. (3) were dropped to obtain the reduced model reported in the results (Table 3).

Uninformative priors were used for all parameters in all models. Specifically, normal distribution with $\mu = 0$ and $\sigma^2 = 10^3$ was used for all parameters (see code in Supplemental Information for implementation). Model parameters were estimated using Markov Chain Monte Carlo (MCMC) simulations with three chains, each with $10^6$ steps with thinning of 100; i.e., $10^4$ values per chain. A burn-in of $2 \cdot 10^3$ steps was used in all cases. All data analyses were conducted in R 3.0.1 (*R Core Team, 2013*); Bayesian analysis was performed using JAGS (*Plummer, 2003*) through rjags package for R (*Plummer, 2013*); coda package for R (*Plummer et al., 2006*) was used to analyse the MCMC output and to perform convergence diagnostics. A complete set of the raw data and the code is available as Supplemental Information. Posterior distributions of all parameter estimates obtained

**Table 3 Parameter estimates describing the dependence of *Sigara striata* mortality on predator traits.** The "full model" contains all predator traits that were measured. Two parameters (a2 and a5) had mean estimated values close to zero. Removing the corresponding terms of the model gave the "reduced model". Explained variance is provided for mean and median parameter values. A model including only predator body mass explained 0.2% of variation in mortality rate.

| Parameter | Full model | | Reduced model | |
|---|---|---|---|---|
| | Mean estimate (SE) | Median (95% CI) | Mean estimate (SE) | Median (95% CI) |
| Intercept (a0) | −3.06 (0.023) | −3.02 (−6.29, −0.08) | −3.35 (0.012) | −3.32 (−5.79, −1.08) |
| Predator mass (linear; a1) | 1.05 (0.006) | 1.02 (0.02, 2.30) | 1.00 (0.004) | 0.98 (0.13, 1.96) |
| Predator mass (quadratic; a2) | −0.19 (0.003) | −0.19 (−0.81, 0.39) | – | – |
| Foraging—ambush (a3) | 2.25 (0.041) | 2.21 (−2.26, 7.03) | 2.38 (0.014) | 2.38 (−0.33, 5.09) |
| Feeding—suctorial (a4) | 2.20 (0.018) | 2.14 (−1.02, 5.81) | 2.15 (0.009) | 2.11 (−0.29, 4.77) |
| Microhabitat—pelagic (a5) | 0.27 (0.036) | 0.30 (−4.45, 4.79) | – | – |
| SD interspecific | 2.24 (0.012) | 2.04 (1.13, 4.46) | 1.89 (0.004) | 1.77 (1.04, 3.42) |
| SD intraspecific | 0.29 (0.001) | 0.25 (0.08, 0.68) | 0.28 (0.001) | 0.25 (0.08, 0.68) |
| Explained variance (%) | 64.1% | 64.3% | 63.2% | 63.3% |

using Gibbs sampling in JAGS are provided in Supplemental Information together with the results of convergence diagnostics and autocorrelation analysis of MCMC chains.

## RESULTS

*Sigara* consumed on average 95% of *Chironomus* midge larvae in the first experiment, while it did not feed on the remaining five species (Fig. 1). Direct observations confirmed that increased mortality of *Chironomus* midge larvae was indeed caused by consumption by *Sigara* and that they were captured alive and subsequently consumed. *Sigara* used its forelegs to grab *Chironomus* larvae on the bottom of the experimental vessel, then ascended to the water surface, where it pierced the cuticle of its prey with a proboscis and sucked out the majority of soft tissues in the *Chironomus* body. Only small part of the prey body was discarded after consumption.

Mortality of *Sigara* caused by individual predators varied widely (Fig. 2). The analysis of the role of predator traits for *Sigara* mortality showed that the mortality of *Sigara* depended on body mass, foraging mode and feeding mode of the predators. On average, larger predators caused higher mortality of *Sigara* (Fig. 3, Table 3). Parameter estimate for the quadratic term of predator body mass was around zero, which indicates that *Sigara* mortality increases in a logistic way (i.e., linearly at the scale of the linear predictor) within the range of predator body masses used in the experiment (Fig. 3, Table 3). Interestingly, body mass played a significant role only when other predator traits were included; alone it explained only ca. 0.2% of variation in mortality rate. In addition to the effect of body mass, the data indicate that predator's foraging mode and feeding mode affect the mortality rate of *Sigara*. Model which included these traits in addition to body mass explained over 60% of variation in mortality rate (Table 3). As expected, ambush predators caused higher mortality of *Sigara* than searching predators (Fig. 3); 8.1 times higher when ambush and searching predators of average body mass are compared. Suctorial predators

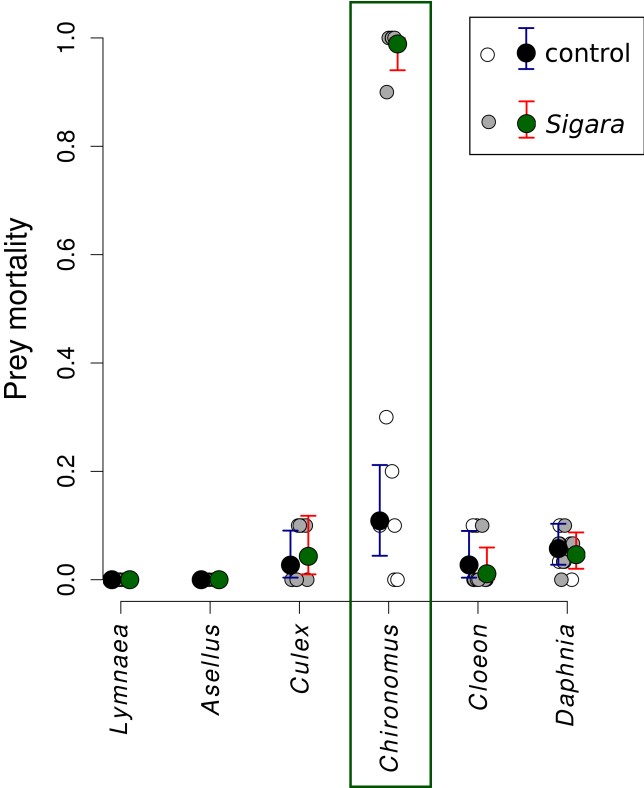

**Figure 1 Mortality of six freshwater invertebrates in the presence/absence of *Sigara striata*.** Estimated proportion of prey individuals dying during a 24 h long experiment. The initial number of prey individuals was six *Lymnaea*, 10 *Chironomus*, 10 *Cloeon*, 10 *Culex*, 10 *Asellus* and 30 *Daphnia*. In the predation treatment, 10 *Sigara* bugs were added. Large circles denote mean values and vertical bars are 95% credible intervals. Small circles show values observed in individual replicates.

killed more individuals than chewing predators (Fig. 3); 6.6 times more when suctorial and chewing predators of average body mass are compared. On the other hand, microhabitat preference of predators had no effect on the mortality of *Sigara* (Table 3).

## DISCUSSION

This study demonstrates that our knowledge of trophic links in freshwater food webs in still insufficient and can be enhanced by detailed laboratory experiments. Food web theory has been attempting to shed light on mechanisms underlying the maintenance of biodiversity (*de Ruiter, Neutel & Moore, 1995*; *McCann, Hastings & Huxel, 1998*) and more recently also to predict the consequences of climate change (*Petchey, Brose & Rall, 2010*; *O'Gorman et al., 2012*), habitat fragmentation (*Melián & Bascompte, 2002*) and other threats to natural communities, as well to study ecosystem recovery (*Layer et al., 2010*). However, empiricists have been frequently critical of food web approaches especially because they have relied on very crude data (e.g., *Polis, 1991*). Early food web studies lumped species to broad functional groups (*Cohen, Briand & Newman, 1990*) and although recent datasets have improved the resolution of food web descriptions considerably (*Thompson, Dunne & Woodward, 2012*), there is still much room for
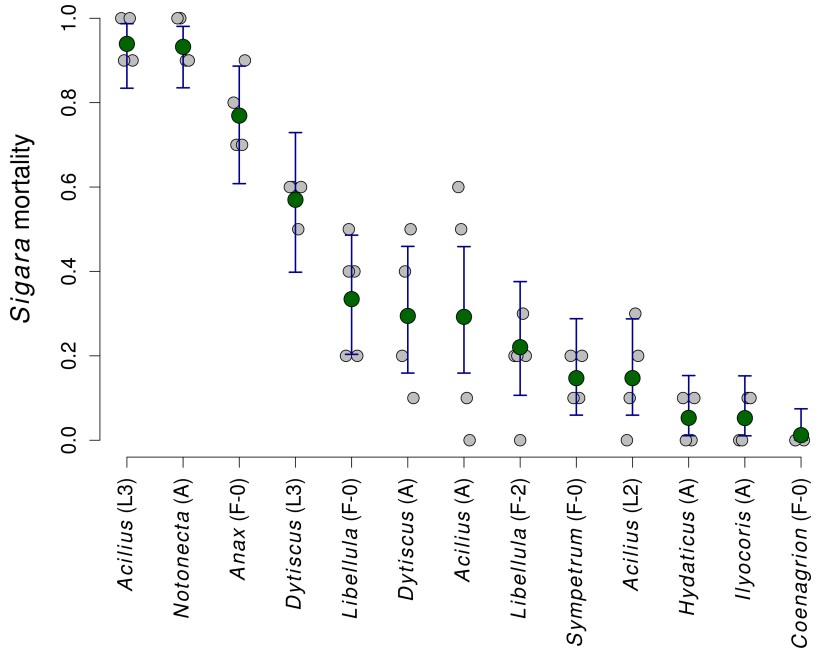

**Figure 2 The mortality of *Sigara striata* caused by 13 different predators.** Estimated proportion of *Sigara* individuals killed by individual predators. Large green circles denote mean values, vertical bars are 95% credible intervals and small grey circles are individual observed values.

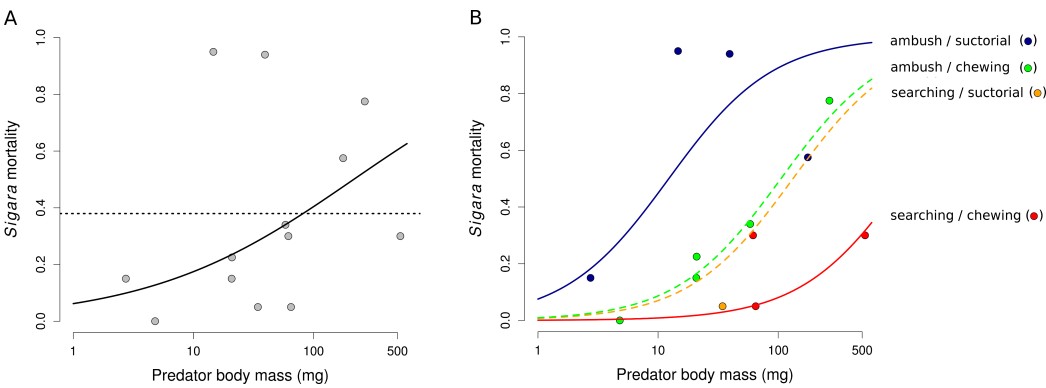

**Figure 3 The dependence of the mortality of *Sigara striata* on predator traits.** Mortality of *Sigara* predicted by a model with predator body mass only (A) and with a combination of body mass, foraging mode and feeding mode (B). Black line in (A) shows model prediction and dotted line shows mean mortality. Lines in (B) show the predicted proportion of *Sigara* killed by a predator as a function of predator body mass, foraging mode and feeding mode. Small circles are mean values of *Sigara* mortality caused by individual predator species. Note that the x-axis has a logarithmic scale.

improvements and refinements. Although food web research has been theory-driven for several decades; changes in the availability of empirical data are responsible for many recent advances. When *May (1972)* reported that complexity decreases stability, he used a simple model where feeding links were assigned at random. It later turned out that non-randomness of feeding links and uneven distribution of interaction strengths allow

complex communities to persist (*de Ruiter, Neutel & Moore, 1995*; *McCann, Hastings & Huxel, 1998*). More detailed data collected over the past two decades led to the rediscovery of the key role of body size for food web structure (*Elton, 1927*; *Brose et al., 2006*), which has important implications for food web stability (*Brose, Williams & Martinez, 2006*; *Heckmann et al., 2012*). New, even more detailed data can reveal additional hidden levels of complexity (*Melián et al., 2011*; *Gilljam et al., 2011*). Detailed observational studies of individual species are thus needed to drive future progress.

Attempts to explain food web structure have recently used body size of predators and prey as a major factor deciding on who eats whom (*Petchey et al., 2008*; *Williams, Anandanadesan & Purves, 2010*; *Williams & Purves, 2011*). However, it is becoming clear that this approach is oversimplified and can be substantially improved by the inclusion of multiple species traits, mostly related to predator foraging behaviour, prey vulnerability, and microhabitat use of both predators and prey (*Rohr et al., 2010*; *Rossberg, Brännström & Dieckmann, 2010*; *Wirtz, 2012*; *Klecka & Boukal, 2013*). Mortality of *Sigara* in my experiment depended not only on predator's body mass, but also on its foraging mode (ambush/searching) and feeding mode (chewing/suctorial). Ambush predators were more efficient in capturing *Sigara* probably because *Sigara* is capable of rapid escape behaviour which may be more effective against searching predators. Suctorial predators also consumed more *Sigara* individuals than chewing predators. Both these results support conclusions reached in multiple-choice experiments with the same set of predators but a larger set of seven different prey species (*Klecka & Boukal, 2013*).

Surprisingly, *Sigara* can also be an important predator in freshwater food webs because the first experiment revealed that it feeds on *Chironomus* midge larvae. The possibility that *Sigara* water bugs could be carnivorous has been debated for several decades (*Popham, Bryant & Savage, 1984*; *Hutchinson, 1993*). Gut content analyses by *Popham, Bryant & Savage (1984)* suggested considerable variation in feeding habits of bugs of the family Corixidae. Several genera, mostly of larger species, such as *Corixa* and *Cymatia*, seem to be mostly carnivorous. However, smaller species of a diverse genus *Sigara* seemed to feed on algae or detritus or on a mixed diet. This reportedly included some unspecified animal components (*Popham, Bryant & Savage, 1984*). Other authors reported remains of microscopic invertebrates, namely rotifers, in the gut of *Sigara* (*Hutchinson, 1993*). Because these bugs are suctorial, only remains of very small organisms which are consumed completely can be found in the gut and majority of the gut contents is unidentifiable (*Popham, Bryant & Savage, 1984*). Sigara in my experiment fed on *Chironomus* midge larvae larger than itself by sucking out their body fluids, while holding the prey using the forelegs. It is very unlikely that any identifiable remains could be detected in its gut. The possibility that *Sigara* could feed on larger invertebrates, including insect larvae, has been rarely considered. For example the review of *Shaalan & Canyon (2009)* on predatory insects feeding on mosquitoes lists only one study reporting predation by *Sigara hoggarica* on mosquitoes (*Alahmed, Alamr & Kheir, 2009*); all other hemipterans in their review were members of the family Notonectidae. Nevertheless, the ability to feed on large prey, even larger than the predator, seems to be common among suctorial predators

(*Klecka & Boukal, 2013*; *Nakazawa, Ohba & Ushio, 2013*). They also kill on average larger amounts of prey than equally sized chewing predators, such as adult beetles or dragonfly larvae, which makes them potentially more likely to have stronger interactions with prey (*Klecka & Boukal, 2013*). Unfortunately, because their diet cannot be reliably studied using gut content analyses, they are underrepresented in studies of food web structure. Given their high species diversity, numerical abundance and voraciousness, this limitation of data availability can significantly distort our understanding of the dynamics of freshwater food webs, especially in small fishless water bodies, where invertebrate predators dominate. It is plausible that *Sigara* is an important freshwater predator, because the abundance of *Sigara* can reach hundreds on ind. m$^{-2}$ (*Bendell & McNicol, 1995*). This makes *Sigara* one of the most abundant insects in suitable habitats (*Tolonen et al., 2003*; *Schilling, Loftin & Huryn, 2009*). Molecular gut content assays have been employed to study the diet of *Notonecta* water bugs (Heteroptera: Notonectidae) in several early studies with mixed results (*Giller, 1982*; *Giller, 1984*; *Giller, 1986*) and in narowly targeted studies to identify predators of mosquitoes (*Morales et al., 2003*; *Ohba et al., 2010*). Although these methods are not widely used, methodical advancements, such as next generation sequencing, hold promise for the future (*Pompanon et al., 2012*). At present, careful laboratory experiments thus remain the only viable option to evaluate the role of the whole group of suctorial predators in freshwater food webs. Supplementing gut content analyses with laboratory predation experiments is a feasible way to boost the reliability and resolution of next generation food web data.

Simple laboratory experiments, such as the ones presented here, are not without limitations. Specifically, they are conducted over short periods of time in small vessels with very simple structure. Short-term measurements of consumption rates may not correlate well with long-term measures of per-capita interaction strength (*Wootton, 1997*). Less clear is how the lack of habitat complexity in laboratory conditions affects prey selectivity of predators. Habitat structure in the natural environment may provide refuges for some prey species against some predators but not against others, as demonstrated in numerous experiments testing the effects of the presence or density of vegetation for the foraging success and selectivity of predatory fish (*Eklöv & Diehl, 1994*; *Horinouchi et al., 2009*). The availability of perching sites also affects foraging behaviour and predation rates in damslefly larvae (*Convey, 1988*). Many other predators adjust their foraging strategies in response to habitat structure; e.g., *Dytiscus* larvae decrease their activity in structured environment and employ more strongly ambushing tactics than in simple environment (*Michel & Adams, 2009*). It is not clear how the relative importance of different predators for the mortality of *Sigara* might differ under natural conditions from the results of my experiment. Importantly, *Sigara* water bugs often live in newly created water bodies lacking dense vegetation, which the simple experimental conditions may represent reasonably well. However, the experimental vessels contained no substrate on the bottom. Many chironomid midge larvae, including the ones used in the experiment, normally bury themselves in soft sediments on the bottom. This is likely to reduce the encounter rates of *Sigara* and *Chironomus* larvae under natural conditions. Estimating the

potential of *Sigara* to exert predation pressure sufficient to control the population density of *Chironomus* in the field will thus require observational or experimental data collected under natural conditions.

The findings of this study, particularly predation by *Sigara* on *Chironomus* larvae and high vulnerability of *Sigara* to ambushing and suctorial predatory insects, have important implications for understanding the structure of freshwater food webs. In suitable habitats, *Sigara* water bugs could be a key resource for certain groups of predators and they may also inflict large predation pressure on chironomid midge larvae, and possibly other invertebrates. It has been recently proposed that *Sigara hoggarica* can regulate mosquito populations by consuming their larvae and pupae (*Alahmed, Alamr & Kheir, 2009*). Although their trophic role is still not well understood, *Sigara* water bugs may thus represent a key group in many freshwater food webs. Another, perhaps trivial, conclusion is that understanding food web structure requires paying attention to details of biology of individual species. In this particular case, information about the possibility that *Sigara* water bugs may play an important role as predators of insect larvae could be gleaned from specialized natural history papers, which are usually not published in high-profile journals and are thus missed by most ecologists. Food web research can progress only if feeding links are assigned correctly and comprehensively with at least species-level resolution. Detailed experimental studies of uncharismatic non-model species may identify a number of unexpected but potentially important trophic links.

## ACKNOWLEDGEMENTS

I am grateful to DJ Wilkinson and R Argiento for introducing me to Bayesian methods during the Applied Bayesian Statistics School in Pavia, Italy, 3.-7.9.2012. I would also like to thank Jan Havelka for providing laboratory space and David S. Boukal for access to equipment needed to conduct this study.

### Funding

This study was supported by "Mattoni Awards for Studies of Biodiversity and Conservation Biology" in 2007–2008 and by Student Grant Agency of the Faculty of Biological Sciences, University of South Bohemia, Czech Republic (SGA 2008). The funders had no role in study design, data collection and analysis, decision to publish, or preparation of the manuscript.

### Grant Disclosures

The following grant information was disclosed by the author:
Mattoni Awards for Studies of Biodiversity and Conservation Biology.
Student Grant Agency of the Faculty of Biological Sciences, University of South Bohemia, Czech Republic: SGA 2008.

## Competing Interests

The author declares there are no competing interests.

## Author Contributions

- Jan Klecka conceived and designed the experiments, performed the experiments, analyzed the data, contributed reagents/materials/analysis tools, wrote the paper, prepared figures and/or tables, reviewed drafts of the paper.

## Supplemental Information

Supplemental information for this article can be found online at http://dx.doi.org/10.7717/peerj.389.

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
