# Peer review of "The role of a water bug, Sigara striata, in freshwater food webs"

_PeerJ, doi:10.7717/peerj.389_

## Round 0.1 · original submission · Minor Revisions

This is a mainly well-written and concise MS in which the author provides good evidence for predatory activity by a water bug and about predation upon the water bug. This type of natural history information is vital to better understanding of ecosystems, and it is only obtained by careful studies of this sort.

Both reviewers recommend minor revisions. And, having read the MS, I can recommend the same.

Both reviewers mention the use of the term "keystone species." The author should seriously consider this particular criticism. One reviewer also mentions some small concerns with the analysis, and in particular that one of the predatory species was misclassified. This should also be addressed.

There are a few other details that the reviewers have pointed out an that the author should attend to as well.

I'd like to thank the author for submitting a good MS to PeerJ, and I'd like to thank both reviewers for their work in assessing this report.

Reviewer 1 ·

Basic reporting

Manuscript Number: 2014:02:1468:0:0:REVIEW

Title: The role of a water bug, Sigara striata, in freshwater food webs: a potential keystone species?

Authors: Jan Klecka

This paper tries to determine that the role of corixid water boatman using two predation experiments in the lab. Results of experiments suggest that Sigara prefers chironomids and is preferred by ambush predator. The statistical approach is very interesting.
This paper dealt with interesting issue concerning competition and may contribute to advance of community ecology of temporary pools. However, the authors should consider some points as below.

The keystone in the title is an exaggerated word for this study because the authors did not estimate the role of Sigara using enclosure and exclosure to impact aquatic community. The authors just estimate their role in the laboratory experiment. Should consider the word “keystone”.

Although I do not understand the Bayesian methods well, I worried about the relationship between variables such as multicollinearity. Does foraging mode and feeding mode of predator have any correlation? In addition, Dytiscus marginalis L3 is suctorial not chewing. Check and reanalyze them as necessary.


P. 3, L. 21: in Klecka and Boukal (2012, 2013).

Experimental design

It's good desgn.

Validity of the findings

These findings are important for aquatic ecology.

Additional comments

Manuscript Number: 2014:02:1468:0:0:REVIEW

Title: The role of a water bug, Sigara striata, in freshwater food webs: a potential keystone species?

Authors: Jan Klecka

This paper tries to determine that the role of corixid water boatman using two predation experiments in the lab. Results of experiments suggest that Sigara prefers chironomids and is preferred by ambush predator. The statistical approach is very interesting.
This paper dealt with interesting issue concerning competition and may contribute to advance of community ecology of temporary pools. However, the authors should consider some points as below.

The keystone in the title is an exaggerated word for this study because the authors did not estimate the role of Sigara using enclosure and exclosure to impact aquatic community. The authors just estimate their role in the laboratory experiment. Should consider the word “keystone”.

Although I do not understand the Bayesian methods well, I worried about the relationship between variables such as multicollinearity. Does foraging mode and feeding mode of predator have any correlation? In addition, Dytiscus marginalis L3 is suctorial not chewing. Check and reanalyze them as necessary.


P. 3, L. 21: in Klecka and Boukal (2012, 2013).

·

Basic reporting

The basic reporting is clear and competent. There are a few locations where articles seem to be missing (e.g., "a", "the"), but these do not affect the clarity of readability of the manuscript.

Experimental design

The experimental design is very simple and reasonable given the stated goals. The research question is clear, and the methods would allow easy reproduction of the study.

Validity of the findings

Although I cannot comment on the validity of the technical details of analyses since I do not use Bayesian statistical methods myself, the fundamental principles of the statistical modeling are sound. The choice of variables and treatment of the results is appropriate. The figures and tables are appropriate and useful, and are clear and legible.

Some content of the Discussion is redundant with the material in the Introduction, but given the short length this is not overbearing. The results and their interpretation are straightforward and the author deals with them appropriately. The importance of the results is appropriately framed and not over-stated. My only suggestion for the Discussion would be to include a section in which the author could speculate or comment on if and how food web relations among Sigara, its prey, and its predators, may differ between this tightly-controlled laboratory experiment and natural habitats.

I also think the invocation of the keystone concept in the title is unwarranted, given that I see no evidence for a keystone role and the concept is not discussed elsewhere in the manuscript. Similarly, in the Abstract there is (reasonable) speculation that Sigara may be less vulnerable to searching predators because it is a fast swimmer, but I don't see a supporting section in the manuscript. I suggest that any results/interpretation in the Abstract should appear in the manuscript itself.

---

## Round 0.2 · accepted · Accept

Thank you once again for submitting your MS to PeerJ. The initial submission was well-written and the reviewers both concurred that the research and interpretation were robust. You have adequately and completely responded to their minor concerns and comments in your revision, and I judge that this MS is now acceptable for publication in PeerJ.